# The Locality and Symmetry of Positional Encodings

**Lihu Chen[1], Gaël Varoquaux[1], Fabian M. Suchanek[2]**
[1] Inria, Soda, Saclay, France
[2] LTCI, Télécom Paris, Institut Polytechnique de Paris, France
{lihu.chen, gael.varoquaux}@inria.fr
{fabian.suchanek}@telecom-paris.fr

## Abstract

Positional Encodings (PEs) are used to inject word-order information into transformer-based language models. While they can significantly enhance the quality of sentence representations, their specific contribution to language models is not fully understood, especially given recent findings that various positional encodings are insensitive to word order. In this work, we conduct a systematic study of positional encodings in **Bidirectional Masked Language Models** (BERT-style) , which complements existing work in three aspects: (1) We uncover the core function of PEs by identifying two common properties, Locality and Symmetry; (2) We show that the two properties are closely correlated with the performances of downstream tasks; (3) We quantify the weakness of current PEs by introducing two new probing tasks, on which current PEs perform poorly. We believe that these results are the basis for developing better PEs for transformer-based language models. The code is available at ⬤ https://github.com/tigerchen52/locality_symmetry

## 1 Introduction

Transformer-based language models with Positional Encodings (PEs) can improve performance considerably across a wide range of natural language understanding tasks. Existing work resort to either fixed (Vaswani et al., 2017; Su et al., 2021; Press et al., 2022) or learned (Shaw et al., 2018; Devlin et al., 2019; Wang et al., 2020) PEs to infuse order information into attention-based models.

To understand how PEs capture word order, prior studies apply visualized (Wang and Chen, 2020) and quantitative analyses (Wang et al., 2021) to various PEs, and their findings conclude that all encodings, both human-designed and learned, exhibit a consistent behavior: First, the position-wise weight matrices show that non-zero values gather on local adjacent positions. Second, the matrices are highly symmetrical, as shown in Figure 1. These

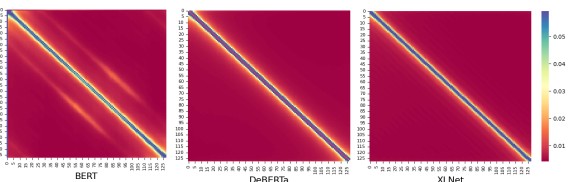

Figure 1: Visualizations of positional weight matrices by using Identical Word Probing (Wang et al., 2021). All matrices highly attend to local positions (*Locality*) and are nearly symmetrical (*Symmetry*).

are intriguing phenomena, with reasons not well understood.

In this work, we focus on uncovering the core properties of PEs in Bidirectional Masked Language Models (BERT-style (Devlin et al., 2019)). We do not include analyses of GPT (Brown et al., 2020) and LLaMA (Touvron et al., 2023) since there is an obvious distinction between them and BERT-style models. Decoder-only models predict the next word based on previous words in the sentence (Left-to-Right mode), and therefore implicitly introduce directional information, which is different from bidirectional models. Furthermore, recent studies have shown that decoder-only transformers without positional encodings are able to achieve competitive or even better performance than other explicit positional encoding methods (Haviv et al., 2022; Kazemnejad et al., 2023). Hence, we focus on the BERT-style models in this work leaving the Decode-only models as a future study.

We study various positional encodings by introducing two quantitative metrics, *Locality* and *Symmetry*. Our empirical studies demonstrate that the two properties are correlated with sentence representation capability. This explains why fixed encodings are designed to satisfy them and learned encodings are favorable to be local and symmetrical. Moreover, we show that if attention-based models are initialized with PEs that already share

good locality and symmetry, they can obtain better inductive bias and significant improvements across 10 downstream tasks. Our findings of the local preference may explain why Sliding Window Attentions (MistralAI, 2023) and Attention Sinks (Xiao et al., 2023) can work effectively.

Although PEs with locality and symmetry can achieve promising results on natural language understanding tasks (such as GLUE (Wang et al., 2019)), the symmetry property itself has an obvious weakness, which is not revealed by previous work. Existing studies use shuffled text to probe the sensitivity of PEs to word orders (Yang et al., 2019a; Pham et al., 2021; Sinha et al., 2021; Gupta et al., 2021; Abdou et al., 2022), and they all assume that the meaning of sentences with random swaps remains unchanged.

However, the random shuffling of words may change the semantics of the original sentence and thus cause the change of labels. For example, the sentence pair below from SNLI (Bowman et al., 2015) satisfies the entailment relation:

a. *A man playing an electric guitar on stage*

b. *A man playing guitar on stage*

If we change the word order of the premise sentence so that it becomes "*an electric guitar playing a man on stage*", a fine-tuned BERT still finds (incorrectly!) that the premise entails the hypothesis.

Starting from this point, we design two new probing tasks of word swap: *Constituency Shuffling* and *Semantic Role Shuffling*. The former preserves the original semantics of the sentence by swapping words inside constituents (local structure) while the latter intentionally changes the semantics by swapping the semantic roles in a sentence (global structure), i.e., the agent and patient. Our results show that existing language models with various PEs are robust against local swaps, but extremely fragile against global swaps. The key contributions of our work are:

- We reveal the core function of PEs by identifying two common properties, Locality and Symmetry, and introduce two quantitative metrics to study them.

- We discover that suitable symmetry and locality lead to better inductive bias, which explains why all positional encodings (both learned or human-designed) exhibit these two properties.

- We design two new probing tasks of word swaps, which show a weakness of existing positional encodings, namely the insensitivity against the swap of semantic roles.

## 2 Preliminaries

The central building block of transformer architectures is the self-attention mechanism (Vaswani et al., 2017). Given an input sentence: $\mathbf{X} = \{\mathbf{x}_1, \mathbf{x}_2, ..., \mathbf{x}_n\} \in \mathbb{R}^{n \times d}$, where $n$ is the number of words and $d$ is the dimension of word embeddings, the attention computes the output of the $i$-th token as:

$$\bar{\mathbf{x}}_i = \sum_{j=1}^{n} \frac{\exp(\alpha_{ij})}{Z} \mathbf{x}_j \mathbf{W}^V \qquad (1)$$

$$\text{where } \alpha_{ij} = \frac{(\mathbf{x}_i \mathbf{W}^Q)(\mathbf{x}_j \mathbf{W}^K)^{\mathsf{T}}}{\sqrt{d}},$$

$$Z = \sum_{j=1}^{n} \exp(\alpha_{ij})$$

Self-attention heads do not intrinsically capture the word order in a sequence because there is no positional constraint in Equation 1. Therefore, specific methods are used to infuse positional information into self-attention (Dufter et al., 2022).

**Absolute Positional Encoding (APE)** computes a positional encoding for each token and adds it to the input content embedding to inject position information into the original sequence. The $\alpha_{i,j}$ in Equation 1 are then written:

$$\alpha_{ij} = \frac{(\mathbf{x}_i + \mathbf{p}_i)\mathbf{W}^Q \big((\mathbf{x}_j + \mathbf{p}_j)\mathbf{W}^K\big)^{\mathsf{T}}}{\sqrt{d}} \qquad (2)$$

Here, $\mathbf{p}_i \in \mathbb{R}^d$ is a position embedding for the $i^{th}$ token, obtained by **fixed** (Vaswani et al., 2017; Dehghani et al., 2019; Takase and Okazaki, 2019; Shiv and Quirk, 2019; Su et al., 2021) or **learned** encodings (Gehring et al., 2017; Devlin et al., 2019; Wang et al., 2020; Press et al., 2021; Ke et al., 2021).

**Relative Positional Encoding (RPE)** produces a vector $\mathbf{r}_{i,j}$ or a scalar value $\beta_{i,j}$ that depends on the relative distance of tokens. Specifically, these methods apply such a vector or bias to the attention head so that the corresponding attentional weight can be updated based on the relative distance of two tokens (Shaw et al., 2018; Raffel et al., 2020):

$$\alpha_{i,j} = \frac{\mathbf{x}_i \mathbf{W}^Q \left(\mathbf{x}_j \mathbf{W}^K + \mathbf{r}_{i,j}^K\right)^\mathsf{T}}{\sqrt{d}}$$

$$\alpha_{i,j} = \frac{(\mathbf{x}_i \mathbf{W}^Q)(\mathbf{x}_j \mathbf{W}^K)^\mathsf{T} + \beta_{i,j}}{\sqrt{d}} \qquad (3)$$

Here, the first mode uses a vector $\mathbf{r}_{i,j}$ while the second uses a scalar value $\beta_{i,j}$, for infusing relative distance into attentional weight. Recent research on RPEs has been remarkably vibrant, with the emergence of diverse novel and promising variants (Dai et al., 2019; He et al., 2021; Press et al., 2022).

**Unified Positional Encoding.** Inspired by TUPE (Transformer with Untied Positional Encoding) (Ke et al., 2021) we rewrite all of the above absolute and relative positional encodings in a unified way as follows:

$$\alpha_{i,j} = \frac{\overbrace{\gamma_{i,j}}^{contextual} + \overbrace{\delta_{i,j}}^{positional}}{\sqrt{d}} \qquad (4)$$

Here, the left half of the numerator, $\gamma_{i,j}$, captures contextual correlations (or weights), i.e., the semantic relations between token $x_i$ and $x_j$. Hence, the contextual correlation can be denoted as $\gamma_{i,j} = (\mathbf{x}_i \mathbf{W}^Q)(\mathbf{x}_j \mathbf{W}^K)^\mathsf{T}$. The right half $\delta$ captures positional correlations, i.e., the positional relations between tokens $x_i$ and $x_j$. For example, the relative encoding in (Shaw et al., 2018) can be represented as $\delta_{i,j} = \mathbf{x}_i \mathbf{W}^Q (\mathbf{r}_{i,j}^K)^\mathsf{T}$. Thus, existing positional encodings all add contextual and positional correlations together in every attention head.

## 3 Positional Encodings Enforce Locality and Symmetry

### 3.1 The Properties of Locality and Symmetry

Existing work analyzes positional encodings with the help of matrix visualizations (Wang and Chen, 2020; Wang et al., 2021; Abdou et al., 2022). Such a matrix is a positional weight map, where each row is a vector for the $i$-th position of the sentence and the element at $(i, j)$ indicates the attention weight between the $i$-th position and the $j$-th position. The matrices are computed by using the *Identical Word Probing* proposed by Wang et al. (2021): many repeated identical words are fed to the pre-trained language model, so that the attention values ($\alpha_{i,j}$ in Equation 4) are unaffected by

contextual weights (We elaborate more on visualizations in Section A.1) The obtained matrices are usually diagonal-heavy, which means that the positional encodings highly attend to local positions. Second, the matrices are usually nearly symmetrical.

We call these two phenomena the *Locality* and *Symmetry* of positional encodings. A prior study shows that the local structure is crucial for understanding the semantics of sentences (Clouatre et al., 2022). The symmetry property has been discovered and quantified already by Wang et al. (2021). Here, we provide a more in-depth analysis of the locality and symmetry. We analyze the linguistic role of locality and point out the potential flaw of symmetry, which is not considered by prior work. To better understand how encodings capture word order, let us consider an attentional weight vector $\epsilon_i$, and the element $\epsilon_{i,j}$ can be denoted as:

$$\epsilon_{i,j} = \frac{\exp(\alpha_{i,j})}{\sum_{j=1}^n \exp(\alpha_{i,j})}$$

$$\text{where } \epsilon_{i,j} \geq 0 \quad \text{and} \quad \sum_{j=1}^n \epsilon_{i,j} = 1 \qquad (5)$$

We then define the two metrics: Locality and Symmetry. **Locality** is a metric that measures how much the weights of an attentional weight vector are gathered in local positions. Given a weight vector for the $i$-th position $\epsilon_i = \{\epsilon_{i,1}, \epsilon_{i,2}, ..., \epsilon_{i,n}\}$, we define locality as:

$$\text{Loc}(\epsilon_i) \in [0,1] = \sum_{j=1}^n \frac{\epsilon_{i,j}}{2^{|i-j|}} \qquad (6)$$

Here, a value of 1 means the vector perfectly satisfies the locality property. For example, given a sequence whose length is 5 and a weight vector for the first position $[1, 0, 0, 0, 0]$, the locality is 1, which means it perfectly matches the locality. In contrast, the locality is $1/16$ if the weight attends only the last position, as in $[0, 0, 0, 0, 1]$. For measuring the locality of a matrix, we average the locality values of all vectors in the matrix.

**Symmetry** is a metric that describes how symmetrical the weights scatter around the current position for an attentional weight vector. To measure the symmetry, we first truncate a weight vector to obtain a new one with the same length to the left and right of the current position $i$: $\epsilon_i^t = \{\epsilon_{i,left}, \epsilon_{i,left+1}, ..., \epsilon_{i,i}, ..., \epsilon_{i,right-1}, \epsilon_{i,right}\}$, where $left$ and $right$ are the left start and the

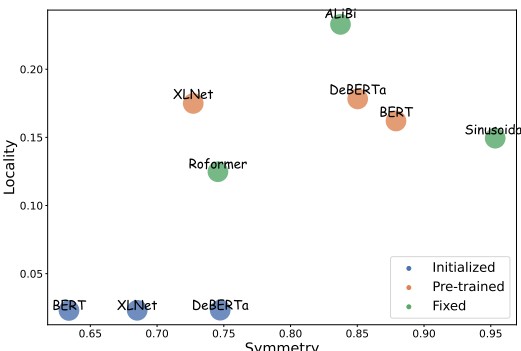

Figure 2: Locality and symmetry values of positional encodings. The green points are fixed and human-designed positional encodings while the orange points are positional encodings after pre-training.

right end position, respectively. The length of the left segment is $len^{left} = i - left$ and the length of the right segment is $len^{right} = right - i$. The two lengths are the same: $len^{left} = len^{right} = \min(i-1, n-i)$. Then, we detect whether the left and right sequences are symmetric with respect to the center position $i$:

$$\text{Sym}(\epsilon_i^t) \in [0,1] =$$
$$1 - \sum_{j=1}^{len^{left}} \frac{\text{Norm}(|\epsilon_{i,j}^t - \epsilon_{i,|\epsilon_i^t|-j+1}^t|)}{len^{left}} \quad (7)$$

First, we apply a min-max normalization to the discrepancy of corresponding position pairs to obtain more uniform distributions. Otherwise, the symmetry values will extremely cluster around 0. Second, we reverse the value so that 1 means a perfect symmetry instead of 0. For example, given a sequence whose length is 5 and a weight vector for the third position $[0.1, 0.2, 0.4, 0.2, 0.1]$, the symmetry is 1, which means that the vector is completely symmetrical.

### 3.2 Are Locality and Symmetry Learned?

To answer this question, we use our two proposed metrics to quantify the locality and symmetry of both manually designed (fixed) and learned encodings. Specifically, three fixed encodings (Sinusoidal (Vaswani et al., 2017), Roformer (or RoPE) (Su et al., 2021), and ALiBi (Press et al., 2022)) and three language models with learned encodings (BERT (Devlin et al., 2019), XLNet (Yang et al., 2019b) and DeBERTa (He et al., 2021)) are ana-

lyzed in this experiment. We measure the two properties of the positional weight matrix (the averaged weight across layers) before and after pre-training.

Figure 2 shows the visualization results. We find that the the three language models all become much more local and symmetrical after pre-training, which proves that the two properties are indeed learned. The manually designed positional encodings, too, are already well localized and symmetrical, which demonstrates that the two properties are important for positional encodings, although the reason is not clear.

### 3.3 Can Locality and Symmetry Yield Better Inductive Bias?

Since locality and symmetry are common features of existing positional encodings, we investigate what happens if a language model is initialized with positional encodings that exhibit good. We conduct empirical studies in both non-pre-trained and pre-trained settings.

**Hand-crafted Positional Encodings** There are various human-designed positional encodings, but the locality and symmetry cannot be modified easily for these encodings. To study the effect of varying locality and symmetry, we design an *Attenuated Encoding* that can be parameterized along these dimensions. Our encoding is defined as follows:

$$\delta_{i,j} = \Phi(l_{i,j}) = \frac{\exp(\alpha_{i,j})}{\sum_{j=1}^n \exp(\alpha_{i,j})}$$
$$\text{where} \quad \alpha_{i,j} = \begin{cases} -s\, w\, l_{i,j}^2 & i \le j \\ -w\, l_{i,j}^2 & i > j \end{cases} \quad (8)$$

Here, $l_{i,j}$ is the relative distance, $w > 0$ is a scalar parameter that controls the locality value, and $s$ is a scalar parameter that controls the symmetry value.

Note that the key difference of our method is that the two properties can be adjusted while other manually designed ones such as the T5 bias (Raffel et al., 2020) and ALiBi (Press et al., 2022) do not allow this.

#### 3.3.1 Non-pre-training Setting

It is impractical to train large language models with different values of locality and symmetry from scratch. Therefore, we use static word embeddings from GloVe (Pennington et al., 2014) and an encoder that is fully based on our handcrafted positional encodings for our experiment. The position-based encoding is adapted from the self-attention

| Model | Size | Sentiment Analysis | | | Textual Entailment | | | Paraphrase Identification | | Textual Similarity | | Avg |
| | | MR (22K) | SUBJ (20K) | SST-2 (68.8K) | QNLI (110K) | RTE (5.5K) | MNLI (413K) | MRPC (5.4K) | QQP (755k) | STS-B (8.4K) | SICK-R (9.4K) | |
| --- | --- | --- | --- | --- | --- | --- | --- | --- | --- | --- | --- | --- |
| BERT | 110M | $72.5_{\pm5.3}$ | $91.0_{\pm2.7}$ | $86.4_{\pm2.7}$ | $85.8_{\pm1.0}$ | $59.2_{\pm1.2}$ | $78.2_{\pm0.8}$ | $73.5_{\pm1.8}$ | $88.7_{\pm0.6}$ | $77.8_{\pm4.1}$ | $64.9_{\pm6.0}$ | 77.8 |
| BERT-$A^*$-$s$ | 113M | $79.4_{\pm2.9}$ | $93.7_{\pm0.6}$ | $88.0_{\pm0.7}$ | $86.3_{\pm1.1}$ | $59.4_{\pm2.7}$ | $78.8_{\pm0.4}$ | $81.5_{\pm2.2}$ | $88.7_{\pm0.4}$ | $83.6_{\pm2.0}$ | $76.3_{\pm1.1}$ | 81.6 |
| BERT-$A^*$ | 138M | $78.2_{\pm3.5}$ | $93.0_{\pm0.8}$ | $88.1_{\pm1.0}$ | $87.0_{\pm0.5}$ | $61.0_{\pm1.4}$ | $78.9_{\pm0.9}$ | $80.9_{\pm3.9}$ | $89.2_{\pm0.3}$ | $84.3_{\pm2.5}$ | $76.0_{\pm4.7}$ | 81.7 |

Table 1: Evaluations of handcrafted encodings across 10 downstream tasks. We report the average score (Spearman correlation for textual similarity and accuracy for others) of five runs using different learning rates. $*$ means the encodings are learnable and $s$ means that positional encodings are shared within the attention headers of layers.

encoder, which means we only keep the $\delta_{i,j}$ in Equation 4. We use our handcrafted attenuated encodings to compute the $\delta_{i,j}$. Therefore, the locality and symmetry can be adjusted easily and we can observe the correlations caused by the changes of the two properties. An implementation example of this positional encoder is shown in Listing 2 of the Appendix A.

We use two sentence-level text classification datasets, MR (Pang and Lee, 2005) and SUBJ (Pang and Lee, 2004), for evaluation. As for the encoder, a single-layer and single-head positional attention is used and the handcrafted encodings are fixed during training. We use the 840B-300d GloVe (Pennington et al., 2014) vectors as word embeddings. For training, we use an Adam optimizer with an initial learning rate 0.002, and introduce a decaying strategy to decrease the learning rate. We adopt a dropout method after the encoder layer, and train models to minimize the cross-entropy with a dropout rate of 0.5. Each model is trained 5 epochs and we select the best model on validation sets to evaluate on the test set. We repeat this procedure 5 times and use the average score to report.

Figure 3 (a) shows the impact of the locality on the performance of the MR dataset. In this experiment, the symmetry value is 1.0 for all encoders. We observe that the accuracy constantly increases as the locality of encodings strengthens, which means a higher locality induces better sentence representation. Experimental results on the SUBJ dataset (Figure A1) show that the accuracy growth slows down at a particular locality value (0.3), which means that a completely perfect locality is unnecessary. The locality value for BERT is around 0.2, and BERT actually does not have an extreme locality. Figure 3 (b) shows the results for different symmetry values. In this experiment, we vary the symmetry while keeping the locality in the interval $[0.15, 0.3]$, which is close to the symme-

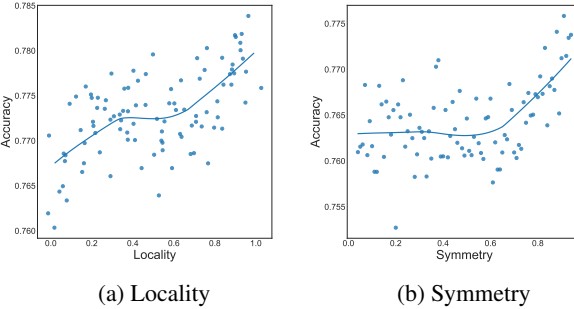

(a) Locality        (b) Symmetry

Figure 3: Empirical studies of the properties of locality and symmetry on the MR sentiment analysis dataset (Pang and Lee, 2005).

try value of BERT. Since the change of symmetry will impact the value of locality, we can only observe this type of partial correlation. We find that symmetry affects performance only after a certain value (0.65), and a larger symmetry leads to better accuracy. However, this does not mean that symmetry is a good property of sentence representations because in many natural language understanding tasks such as sentiment analysis do not require strict word order information. In section 3.5, we will discuss a potential flaw caused by symmetry. Experiments on the SUBJ dataset (Pang and Lee, 2004) lead to similar conclusions, and are shown in Figure A1 in the Appendix A.

### 3.3.2 Pre-training Setting

In this experiment, we adjust the parameters $w$ and $s$ in Equation 8 to obtain a weight vector $\delta$ that shares the locality and symmetry of the pre-trained BERT (Locality=0.17 and Symmetry=1.0). We pre-train BERT$_{base}$ initialized with $\delta$ and compare them to learned encodings on downstream natural language understanding tasks. Two variants are compared with the original BERT: 1) BERT-$A^*$-$s$ uses learnable and shared $\delta$, but the weights are shared inside a particular layer; 2) BERT-$A^*$ uses learnable but not shared $\delta$, which means $\delta$

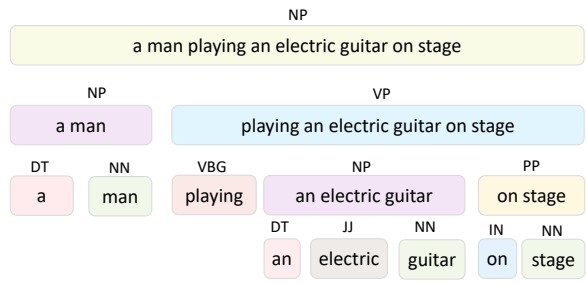

Figure 4: Illustration of constituent parsing for one sentence in SNLI "*a man playing an electric guitar on stage*", generated by Berkeley Neural Parser (Kitaev et al., 2019).

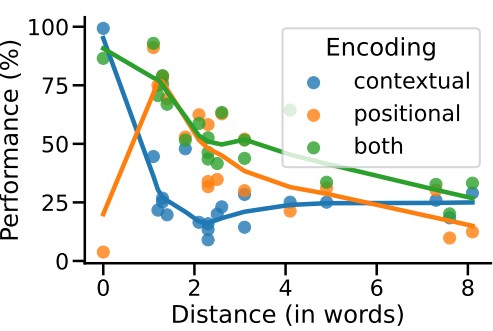

Figure 5: Accuracy of top-20 dependency relations. Detailed results in Table A3, Appendix.

is different in each attentional head. More details of the datasets and pre-training are shown in Appendix A.4 and Section A.2, respectively. The empirical results are shown in Table 1. We observe that both BERT-$A^*$-$s$ and BERT-$A^*$ can significantly outperform the original BERT, which demonstrates positional encodings with initialization of suitable locality and symmetry can have a better inductive bias in sentence representation. The visualizations of attentional heads in BERT are shown in Figure A3 and the visualizations of $\delta$ in BERT-$A^*$ are shown in Figure A4, both in the AppendixA. In fact, there is a great diversity of behaviors within different attentional heads, mainly for the diagonal bandwidths of attentional maps (locality), which signifies proximity units can be combined at different distances.

We conclude that positional encodings with more suitable locality and symmetry can yield better performance on downstream tasks, which may explain why fixed encodings are designed to meet the two properties and why learned encodings all exhibit this behavior. However, learned encodings are not perfectly local, but they still have the ability to represent sentences very well, which might be due to the network architectures and the specific target tasks. Moreover, increasing the diversity of positional correlations in different attentional heads, e.g., the diversity of locality, is also beneficial for representation capabilities.

### 3.4 Why the Locality Matters?

*Locality* means that the positional weights favor the combination of units in a sentence to their adjacent units when creating higher-level representations. For example, sub-tokens can be composed into lexical meanings (e.g., { *"context"*, *"##ual"*} → *"contextual"*) or words can be composed into

phrase-level meaning (e.g., {*"take"*, *"off"*} → *"take off"*), and clause-level and sentence-level meaning can be obtained through an iterative combination of low-level meanings, which is consistent with the multi-layer structure in pre-trained language models. From a linguistic perspective, words linked in a syntactic dependency should be close in linear order, which forms what can be called a dependency locality (Futrell et al., 2020). Dependency locality provides a potential explanation for the formal features of natural language word order. Consider the two sentences *"John throws out the trash"* and *"John throws the trash out"*. Both are grammatically correct. There is a dependency relationship between *"throws"* and *"out"* and the verb is modified by the adverb. However, language users prefer the expression with the first sentence because it has a shorter total dependency length (Dyer, 2017; Liu et al., 2017; Temperley and Gildea, 2018). Based on the visualizations and dependency locality, we, therefore, speculate that one main function that positional encodings have learned during pre-training is local composition, which exists naturally in our understanding of sentences. Empirical studies also demonstrate that performances of shuffled language models are correlated with the violation of local structure (Khandelwal et al., 2018; Clouatre et al., 2022).

To verify that the main function of locality is to compose adjacent units with short-distance dependency relations, we examine the dependency knowledge stored in positional weights. Specifically, we follow the syntactic probing test by Clark et al. (2019), and then each head in PLMs is regarded as a simple predictor of dependency relations. Given the attention weight vector of an input word, we output the word with the highest values and think the pair of words holds some type of

| | Original | Shuffled |
|---|---|---|
| Shuffled-3 | *An old man* *with a package* *poses in front* *of an advertisement* . | *An man old* *with package a* *poses in front* *of advertisement an* . |
| Shuffled-4 | A land rover is being driven across a river . | A land rover is being a driven river across . |
| Shuffled-5 | A man reads the paper in a bar with green lighting . | A man reads the paper in with green a lighting bar . |
| Shuffled-6 | A little boy in a gray and white striped sweater and tan pants is playing on a piece of playground equipment . | A little boy in striped a sweater and white gray and tan pants is playing piece playground of equipment on a . |
| Shuffled-SR | several women are playing volleyball . | volleyball are playing several women . |
| Shuffled-SR | a man and woman are sharing a hotdog . | a hotdog are sharing a man and woman . |

Table 2: Some cases of the shuffled SNLI datasets in our word swap probing. Shared color indicates corresponding phrases.

| Model | Symmetry | Locality | Original | Shuffle-3 ($\Delta$) | Shuffle-4 ($\Delta$) | Shuffle-5 ($\Delta$) | Random ($\Delta$) | Original | Shuffle-SR ($\Delta$) |
|---|---|---|---|---|---|---|---|---|---|
| BERT | 87.9 | 16.2 | 89.8 | -0.4 | -0.6 | -0.3 | -2.7 | 89.8 | -63.9 |
| ALBERT | 82.0 | 20.3 | 91.8 | -0.5 | -1.1 | -1.3 | -6.0 | 92.0 | -66.8 |
| DeBERTa | 85.0 | 17.8 | 91.6 | -0.5 | -0.7 | -1.3 | -5.1 | 91.6 | -58.9 |
| XLNet | 72.7 | 17.5 | 91.5 | -0.2 | -0.3 | -0.7 | -5.4 | 91.3 | -57.8 |
| StrucBERT | 96.3 | 7.5 | 90.9 | -0.5 | -0.9 | -1.3 | -4.4 | 90.8 | -44.6 |

Table 3: Results of Constituency Shuffling and Semantic Role Shuffling, measured by accuracy. Shuffle-$x$ means phrases with length $x$ are shuffled. Shuffle-SR means the semantic roles of the agent and patient are swapped.

dependency relation. While no single head can perform well on all relations, the best-performed head is selected as the final ability of a model for each particular relation. In this experiment, we adopt the original TUPE (Ke et al., 2021) that uses absolute positional encodings as our base model. The ability of contextual and positional weights is evaluated by setting the unrelated correlations to zero (in Eq 4), e.g., the first term is set to dumb when checking the ability of positional weights. The two variants are referred as to *contextual* and *positional* attention, respectively.

We extract attention maps from BERT on the MRPC (Dolan and Brockett, 2005) annotated by the dependency parser of spaCy [1]. We report the results on top-20 dependency relations.

Figure 5 shows performance on relations with different distances. (Table A3 gives relation-specific results). First, we observe that positional attention is significantly more important than contextual attention in short-distance dependency relations (distance from 1 to 4). Second, contextual attention takes the lead on long-distance relations (after 6). Again, the combination of the two features can yield the best performance. The "outlier" in the lower left corner is the Root dependency. Because this relation is a self-reflexive edge, contextual (or self) attentions can performs well on

it while learned PEs do not attend to the current word itself, e.g., visualizations of BERT and DeBERTa in Figure 1. Moreover, our empirical results show that there is a clear distinct role between the positional and contextual encodings in sentence comprehension: positional encodings play more of a role at the syntactic level while contextual encodings serve more at the semantic level (see Section A.6.1 in the Appendix A).

We summarize that the locality property guides positional encodings to capture more short-distance dependencies while contextual weights capture more long-distance ones.

### 3.5 What Is the Drawback of Symmetry?

Although positional encodings with good symmetry perform well on a series of downstream tasks, the symmetry property has an obvious flaw in sentence representations, which is ignored by prior studies.

The *symmetry* (also observed by Wang and Chen (2020); Wang et al. (2021)) of the positional matrices implies that the contributions of forward and backward sequences are equal when combining adjacent units under the locality constraint. This is contrary to our intuition, as the forward and backward tokens play different roles in the grammar, as we have seen in the examples of "*a man playing an electric guitar on stage*" and "*an electric guitar playing a man on stage*". However, this symmetry

---

[1] https://spacy.io/api/dependencyparser

is less disruptive at the local level inside sentences. Recent work in psycholinguistics has shown that sentence processing mechanisms are well designed for coping with word swaps (Ferreira et al., 2002; Levy, 2008; Gibson et al., 2013; Traxler, 2014). Further, Mollica et al. (2020) hypothesizes that the composition process is robust to local word violations. Consider the following example:

a. *on their last day they were overwhelmed by farewell messages and gifts*

b. *on their last day they were overwhelmed by farewell and messages gifts*

c. *on their last they day were overwhelmed farewell messages by and gifts*

The local word swaps (colored underlined words) are introduced in the latter two sentences, leading to a less syntactically well-formed structure. However, experimental results show that the neural response (fMRI blood oxygen level-dependent) in the language region does not decrease when dealing with word order degradation (Mollica et al., 2020), suggesting that human sentence understanding is robust to local word swaps. Likewise, symmetry can be understood in this way: when a reader processes a word in a sentence, the forward and backward nearby words are the most combinable, and the comprehension of this composition is robust to its inside order. On the other hand, symmetry is not an ideal property for sentence representations (consider the case of "*an electric guitar*"). Next, we use two new probing tasks of word swap to illustrate the flaws of symmetry.

Existing probes study the sensitivity of language models to word order by shuffling the words in a sentence, and they can be roughly divided into three categories: random swap (Pham et al., 2021; Gupta et al., 2021; Abdou et al., 2022), n-gram swap (Sinha et al., 2021), and subword-level swap (Clouatre et al., 2022). All these studies assume that the labels of the randomly shuffled sentences are unchanged. However, this is obviously not the case. In particular, the shuffled sentence may have another label (think of the textual entailment example from the introduction).

To address the issue, we propose two new probing tasks of word swaps: *Constituency Shuffling* and *Semantic Role Shuffling*. *Constituency Shuffling* aims to disrupt the inside order of constituents, which is able to change the word order while preserving the maximum degree of original semantics.

As an example, consider the constituent parsing in Figure 4. We can easily shuffle the word order inside "*an electric guitar*" (say, to, "*guitar an electric*"), which will lead to a grammatically incorrect, but still comprehensible sentence because humans are able to understand sentences with local word swaps (Ferreira et al., 2002; Levy, 2008). To construct such shuffled datasets, the premise sentences in the SNLI (Bowman et al., 2015) test set are shuffled and we keep the hypothesis sentences intact. Here, we let $x \in [3, 5]$ and select a subset from SNLI to make sure that every premise sentence has at least one phrase with a length from 2 to 5. We select five types of target phrases for shuffling: *Noun Phrase, Verb Phrase, Prepositional Phrase, Adverb Phrase, and Adjective Phrase*. Finally, a Shuffle-$x$ SNLI is obtained by disrupting the order inside a phrase with length $x$ and the size for each shuffle-$x$ is around 5000.

Our other task, *Semantic Role Shuffling*, intentionally changes the semantics of the sentences by swapping the order of the agent and patient of sentences. For example, in Figure 4, "*a man*" is the entity that performs the action, technically known as the agent, and "*an electric guitar*" as the entity that is involved in or affected by the action, which is called the patient. Our dataset Shuffle-SR swaps these semantic roles. Some examples are shown in Table 2.

To probe the sensitivity of language models to the two types of shuffling, we fine-tune 5 pre-trained language models with good symmetry on the SNLI training set and evaluate them on the newly constructed Shuffle-$x$ and Shuffle-SR datasets (see details in Appendix A.3). The overall results of the word swap probing are shown in Table 3. We first observe that the performances of all language models across Shuffle-$x$ datasets (local swaps) basically do not degenerate, which confirms the benefits of the locality and symmetry properties. Second, most models fail on the Shuffle-SR dataset (global swaps), which demonstrates local symmetry does not capture global swaps well. This explains the reason that BERT-style models fail on the example: "*an electric guitar playing a man on stage*". Although the local symmetry learned by positional encodings can perform well on a series of language understanding tasks, the symmetry itself has obvious flaws. The better performance of StrucBERT on the Shuffle-SR suggests that introducing additional order-sensitive training tasks

may improve this problem. More details of the probing tasks are described in Appendix A.3.

# 4 Conclusion

We have proposed a series of probing analyses for understanding the role of positional encodings in sentence representations. We find two main properties of existing encodings, Locality and Symmetry, which are correlated with the performance of downstream tasks. We first investigate the linguistic role of positional encodings in sentence representation. Meanwhile, we point out an obvious flaw of the symmetry property. We hope that these findings will inspire future work to better design positional encodings.

# Limitations

The limitations of this work are three-fold. First, our study focuses on positional encoding in Masked Language Models (bidirectional), but this work does not involve decoding-only language models (GPT-style). Furthermore, recent studies have shown that decoder-only transformers without positional encodings is able to outperform other explicit positional encoding methods (Haviv et al., 2022; Kazemnejad et al., 2023), which deserves further research. Second, our analysis is limited to the natural language understanding of the English language. Different languages display different word order properties. For instance, English is subject-verb-object order (SVO) while Japanese is subject-object-verb order (SOV), and natural language generation tasks are not included in this work. Third, although our handcrafted positional encodings satisfy the symmetry property, they merely replicate the limitations of current positional encoding, albeit in a simplified form. Further architecture development should address the problem of the "*an electric guitar playing a man on stage.*" mentioned in the introduction.

# Acknowledgements

This work was partially funded by projects NoRDF (ANR-20-CHIA-0012-01) and LearnI (ANR-20-CHIA-0026).

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

## A  Appendix

### A.1  Visualizations of Positional Encodings

To understand what positional encodings learn after pre-training, we visualize the positional weights in attentional heads. The Identical Word Probing is adopted in this experiment (Wang et al., 2021). The used pre-trained language models are shown in Table A1, and the repeated words are randomly selected from the corresponding vocabulary. Note that sub-tokens like single characters and non-physical words are removed. For visualization, we adopt the Identical Word Probing proposed by Wang et al. (2021), which feeds many repeated identical words to pre-trained language models and thus the attention values are disentangled with contextual weights. More specifically, we randomly select 100 words from the corresponding vocabulary (filtering out single characters and sub-words such as "##nd"). We repeat each word to compose a sentence of length 128. These 100 sentences are fed into a language model and the attention weights across different layers are averaged as the positional weight matrix of a particular language model.

### A.2  Details of Pre-training

We use the configuration of the original $BERT_{base}$ (Devlin et al., 2019) with 110M parameters for pre-training. Our model is implemented with PyTorch using the pytorchic-bert tool[2]. The number of layers, attention heads, and the projection dimension are 12, 12, and 768 respectively. We use the original vocabulary with a size of 30522. The training corpus is the English Wikipedia (20200101 dumps), which totals 13G after preprocessing by WikiExtractor. We pre-train with sequences of at most $T = 512$ tokens and set the batch size as 64 to pre-training 600K steps. The optimizer is Adam with a learning rate of 5e-4, $\beta1 = 0.9$, $\beta2 = 0.999$, L2 weight decay of 0.01, and a warmup rate of 0.1. The dropout probability is always set as 0.1.

---

[2]https://github.com/dhlee347/pytorchic-bert

We use the original $BERT_{base}$ as our backbone and vary the positional encodings to pre-training different variants for comparison. Listing 1 shows a code example about how to inject handcrafted positional encodings into the BERT backbone. Each variant is fine-tuned on the training dataset with different learning rates (among 9e-5, 7e-5, 5e-5, 3e-5, 1e-5). After, we evaluate the fine-tuned model on the Dev set and report the average score of five learning rates. Apart from BERT, we introduce the TUPE model as another baseline. Specifically, we pre-train the following variants:

```python
class MultiHeadedSelfAttention(nn.Module):
    """ Multi-Headed Scaled Dot Product Attention """
    def __init__(self, config):
        super().__init__()
        self.n_heads = config.n_heads
        self.drop = nn.Dropout(config.p_drop_attn)
        self.proj_q = nn.Linear(config.dim, config.dim)
        self.proj_k = nn.Linear(config.dim, config.dim)
        self.proj_v = nn.Linear(config.dim, config.dim)

    def forward(self, x, mask, pe):
        """
        x, q(query), k(key), v(value) : (B(batch_size),
            S(seq_len), D(dim))
        mask : (B(batch_size) x S(seq_len))
        pe: positional weights (B(batch_size), H(Head_number)),
            S(seq_len), S(seq_len))
        * split D(dim) into (H(n_heads), W(width of head)) ; D
            = H * W
        """
        # (B, S, D) -proj-> (B, S, D) -split-> (B, S, H, W)
            -trans-> (B, H, S, W)
        q, k, v = self.proj_q(x), self.proj_k(x), self.proj_v(x)
        q, k, v = (split_last(x, (self.n_heads,
            -1)).transpose(1, 2)
        for x in [q, k, v])
        # (B, H, S, W) @ (B, H, W, S) -> (B, H, S, S)
            -softmax-> (B, H, S, S)
        scores = q @ k.transpose(-2, -1) / np.sqrt(k.size(-1))

        # inject positional weights into contextual weights
        # (B, H, S, S) + (B, H, S, S) -> (B, H, S, S)
        scores = scores + pe

        if mask is not None:
        mask = mask[:, None, None, :].float()
        scores -= 10000.0 * (1.0 - mask)

        scores = self.drop(F.softmax(scores, dim=-1))
        # (B, H, S, S) @ (B, H, S, W) -> (B, H, S, W) -trans->
            (B, S, H, W)
        h = (scores @ v).transpose(1, 2).contiguous()
        # -merge-> (B, S, D)
        h = merge_last(h, 2)
        return h
```

Listing 1: A code example of how to inject handcrafted positional encodings into self-attentions.

- BERT is the original one and we use it as a baseline.

- BERT-$A^*$ is a variant of the former, but the encodings are learnable during pre-training.

- BERT-$A^*$-$s$ shares learnable positional encodings within a layer.

Suppose that the hidden dimension is 768, the layer number is 12, the head number is 12, and the maximum length is 512 for $BERT_{base}$ model, we can

| Model | Size | Version | Language |
|-------|------|---------|----------|
| BERT | 110M | *bert-base-uncased* | English |
| DeBERTa | 100M | *microsoft/deberta-base* | English |
| XLNet | 110M | *xlnet-base-cased* | English |

Table A1: Details of pre-trained language models used in visualizations.

| Model | Size | Version | Fine-tuned by us |
|-------|------|---------|------------------|
| BERT | 110M | *bert-base-uncased* | ✓ |
| ALBERT | 223M | *ynie/albert-xxlarge-v2-snli_mnli_fever_anli_R1_R2_R3nli* | ✗ |
| DeBERTa | 100M | *microsoft/deberta-base* | ✓ |
| XLNet | 340M | *ynie/xlnet-large-cased-snli_mnli_fever_anli_R1_R2_R3-nli* | ✗ |
| StrucBERT | 340M | *bayartsogt/structbert-large* | ✓ |

Table A2: Details of pre-trained language models used in word swap probing.

calculate the size for each variant. The number of parameters of handcrafted positional encoding for each head is 262K ($512 \times 512$). If positional heads are different across all layers, the total cost is 37.7M ($512 \times 512 \times 12 \times 12$). If the positional encodings are shared across all attentional heads, the total cost is 3.1M ($512 \times 512 \times 12$).

### A.3 Word Swap Probing

To validate if language models with positional encodings are sensitive to the local and global word swaps, we construct Shuffle-$x$ and Shuffle-SR SNLI datasets. Shuffle-$x$ means the word orders of phrases with length $x$ are disrupted, e.g. "*an electric guitar*" is a 3-gram phrase, and it might be "*guitar an electric*" in Shuffle-3 SNLI. In this way, a new sentence with the same meaning can be obtained and therefore the initial label of the sample will not be changed. To construct such shuffled datasets, the premise sentences in the SNLI test set are shuffled and we keep the hypothesis sentences intact. Here, we let $x \in [3, 5]$ and select a subset from SNLI to make sure that every premise sentence has at least one phrase with length from 2 to 5. We select five types of target phrases for shuffling: *Noun Phrase, Verb Phrase, Prepositional Phrase, Adverb Phrase, and Adjective Phrase*. Finally, a Shuffle-$x$ SNLI is obtained by disrupting the order inside a phrase with length $x$ and the size for each shuffle-$x$ is around 5000. The first fourth rows in Table 2 shows some samples.

As for the Shuffle-SR SNLI dataset, the semantic roles of the agent and patient are swapped in a sentence. We use the Algorithm 1 to collect a

subset from the SNLI test set. This algorithm is applied successively to the premise and hypothesis sentence for a sample whose label is entailment, and if the result of either of them is not null, we consider it a valid shuffled sample, which means we only shuffle the premise or hypothesis. After, we can obtain a new sample and the pair of sentences are contradicted with each other. In total, there are 1329 samples. To ensure that all sentences are semantically correct, we manually selected 300 pairs from them. The last two rows in Table 2 show two examples in the Shuffle-SR dataset.

To probe the capabilities of language models on our newly constructed datasets, we adopt five different pre-trained language models (as shown in Table A2) and we use Hugging Face for implementation (Wolf et al., 2020). These models are fine-tuned on the training set of SNLI, and the model with the best score on the validation set is stored for the following experiments. Note that there are off-the-shell ALBERT and XLNet for natural language inference, we therefore use them directly without fine-tuning. During the fine-tuning stage, the maximum length of the tokenized input sentence pair is 128, and the optimizer is Adam (Kingma and Ba, 2015) with a learning rate of 2e-5. The batch size is 32 and the epoch is 3. After fine-tuning, the best model is evaluated on our shuffle SNLI test set, and we record their performances when faced with local and global word swaps.

### A.4 Details of Downstream Datasets

SentEval is based on a set of existing text clas-

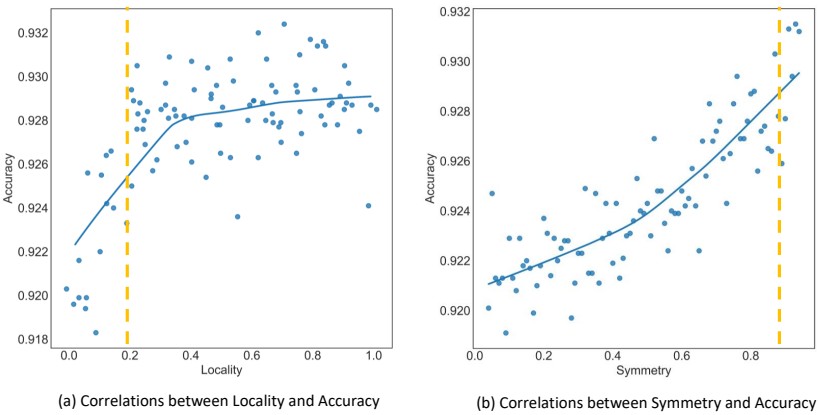

(a) Correlations between Locality and Accuracy

(b) Correlations between Symmetry and Accuracy

Figure A1: Correlations between the two properties (Locality and Symmetry) and accuracy on SUBJ dataset (Pang and Lee, 2004). The yellow line shows the locality or symmetry of the pre-trained BERT.

---

**Algorithm 1:** Construction of Shuffle-SR Sentences

**Input:**
$s$: a premise or hypothesis sentence in SNLI,
$\mathcal{A}$: Auxiliary verb list
$\mathcal{M}$: Semantic Role Labeling Model,
$\mathcal{D}$: Subject and Object Case Mapping
// e.g., I ↔ me

**Output:** A sentence $s^*$ with shuffled agent and patient or *None*

1   $\mathcal{R} \leftarrow$ Predict the semantic roles of words in sentence $s$ by using the model $\mathcal{M}$
2   $\mathcal{V} \leftarrow$ Take the verb list from $\mathcal{R}$
3   **foreach** *verb $v$ in $\mathcal{V}$* **do**
4      **if** *$v$ appears in $\mathcal{A}$* **then**
5         continue
6      **if** *$\mathcal{R}$ does not contain an agent or patient* **then**
7         continue
8      $a, p \leftarrow$ Take the agent and patient from $\mathcal{R}$
9      $s^* \leftarrow$ Swap the $a, p$ in sentence $s$
10     $s^* \leftarrow$ Transform the subject and object case in $s^*$ if $a$ or $p$ in $\mathcal{D}$
11     **return** $s^*$
12 **return** *None*

---

sification tasks involving one or two sentences as input. However, most tasks in SentEval are closely related to sentiment analysis and thus not diverse enough. GLUE benchmark introduces a series of difficult natural language understanding tasks while some particular tasks only contain one dataset, e.g., sentiment analysis and textual similarity. Moreover, the size of WNLI in GLUE is rather small and the GLUE webpage notes that there are issues with the construction of this dataset [3]. To better evaluate the capability of models for sentence representation, we, therefore, select 10 datasets from SentEval and GLUE, covering four types of sentence-level tasks:

- **Sentiment Analysis** is also known as opinion mining, which aims to classify the polarity of a given text, whether the expressed opinion is positive, negative, or neutral. We use MR (Pang and Lee, 2005), SUBJ (Pang and Lee, 2004), and SST (Socher et al., 2013) for this task.

- **Textual Entailment** describes the inference relation between a pair of sentences, whether the premise sentence entails the hypothesis sentence. Actually, this is a classification task with three labels: entailment, contradiction, and neutral. Here, we use QNLI (Rajpurkar et al., 2016), RTE (Dagan et al., 2005; Haim et al., 2006; Giampiccolo et al., 2007; Bentivogli et al., 2009) and MNLI (Williams et al., 2018) for evaluation. Note that we report the average score for the two test sets of MNLI.

- **Paraphrase Identification** is to determine whether a pair of sentences have the same meaning. We use MRPC (Dolan and Brockett, 2005) and QQP ( data.quora.com/ First-Quora-Dataset-Release-Question-Pairs) for evaluation.

---

[3] https://gluebenchmark.com/faq

- **Textual Similarity** deals with determining how similar two pieces of text are. We use STS-B (Cer et al., 2017) and SICK-R (Marelli et al., 2014) for evaluation.

## A.5 Details of TUPE Model

In absolute positional encoding, the positional encoding is added together with the contextual encoding:

$$\alpha_{ij} = \frac{(\mathbf{x}_i + \mathbf{p}_i)\mathbf{W}^Q\big((\mathbf{x}_j + \mathbf{p}_j)\mathbf{W}^K\big)^\mathsf{T}}{\sqrt{d}} \quad \text{(A.1)}$$

where $\mathbf{p}_i \in \mathbb{R}^d$ is a position embedding of the $i$-th token. Further, the above equation can be expanded as:

$$\alpha_{ij} = \frac{(\mathbf{x}_i\mathbf{W}^Q)(\mathbf{x}_j\mathbf{W}^K)^\mathsf{T}}{\sqrt{d}} + \frac{(\mathbf{x}_i\mathbf{W}^Q)(\mathbf{p}_j\mathbf{W}^K)^\mathsf{T}}{\sqrt{d}}$$
$$+ \frac{(\mathbf{p}_i\mathbf{W}^Q)(\mathbf{x}_j\mathbf{W}^K)^\mathsf{T}}{\sqrt{d}} + \frac{(\mathbf{p}_i\mathbf{W}^Q)(\mathbf{p}_j\mathbf{W}^K)^\mathsf{T}}{\sqrt{d}}$$
$$\text{(A.2)}$$

There are four terms in this expression: context-to-context, context-to-position, position-to-context, and position-to-position. While the first and the fourth term are intuitive, the token encodings and positional encodings do not have strong correlations with each other, and the context-position correlations may even induce unnecessary noise. Based on this analysis, Ke et al. (2021) propose TUPE (Transformer with Untied Positional Encoding) that removes the second and third redundant terms and introduces different parameters for the position encoding:

$$\alpha_{ij} = \frac{(\mathbf{x}_i\mathbf{W}^Q)(\mathbf{x}_j\mathbf{W}^K)^\mathsf{T} + (\mathbf{p}_i\mathbf{U}^Q)(\mathbf{p}_j\mathbf{U}^K)^\mathsf{T}}{\sqrt{d}},$$
$$\text{(A.3)}$$

Here, $\mathbf{U}^Q$ and $\mathbf{U}^K$ are weights that need to be learned, capturing positional queries and keys, respectively. Their empirical results confirm that the removal of the two context-to-position terms consistently improves the model performance on a series of tasks.

```python
class MultiHeadPositionalAttention(nn.Module):
    """ Multi-Headed Scaled Dot Product Attention """
    def __init__(self, config):
        super().__init__()
        self.n_heads = config.n_heads
        self.drop = nn.Dropout(config.p_drop_attn)

    def forward(self, x, mask, pe):
        """
        x, q(query), k(key), v(value) : (B(batch_size),
            S(seq_len), D(dim))
        mask : (B(batch_size) x S(seq_len))
        pe: positional weights (B(batch_size),
            H(Head_number)), S(seq_len), S(seq_len))
        * split D(dim) into (H(n_heads), W(width of head)) ;
            D = H * W
        """
        # (B, S, D) -proj-> (B, S, D) -split-> (B, S, H, W)
            -trans-> (B, H, S, W)
        q, k, v = (split_last(x, (self.n_heads,
            -1)).transpose(1, 2)
        for x in [q, k, v])
        # (B, H, S, W) @ (B, H, W, S) -> (B, H, S, S)
            -softmax-> (B, H, S, S)

        scores = pe
        if mask is not None:
            scores.masked_fill_(~mask, 0.)

        # (B, H, S, S) @ (B, H, S, W) -> (B, H, S, W)
            -trans-> (B, S, H, W)
        h = (scores @ v).transpose(1, 2).contiguous()
        # -merge-> (B, S, D)
        h = merge_last(h, 2)
        return h
```

Listing 2: A code example of the Positional Attention.

## A.6 Linguistic Roles of Positional Ecnodings

Positional and contextual weights are usually entangled in every attentional head, and therefore the behavior of positional encodings cannot be observed independently (as shown in Equation 4).

To address this, we set the contextual correlation $\gamma_{i,j}$ (in Equation 4) to zero (instead of removing the contextual encodings completely) and thus the attentional weight $\alpha_{i,j}$ only depends on positional correlation. Note that this operation does not alter the structure of the original network because a softmax layer is applied to the vector $\alpha_{\mathbf{i}}$, and the output is still an attentional weight vector that can be regarded as a kind of discrete probability distribution. Therefore, the output sentence representation is decoupled from contextual encoding. We refer to this adapted model as BERT-$p$. For comparison, we remove the positional correlations $\delta_{i,j}$ to obtain BERT-$c$. We do the same for a pre-trained TUPE model, to obtain TUPE-$p$ and TUPE-$c$, and we already described TUPE in Section A.5.

### A.6.1 Linguistic Probing Tasks

In this linguistic probing, we adopt widely used 10 probing tasks (Conneau et al., 2018) with a standard evaluation toolkit (Conneau and Kiela, 2018). The following are the details of each probing task, including three categories:

- SentLen (Surface) aims to predict the length of sentences in terms of the number of words, and the dataset is constructed following Adi et al. (2017).

- WC (Surface) means word content, which checks whether it is possible to recover information about the original word from the embedding of the sentence.

- BShift (Syntactic) means bigram shift. In this task, two random adjacent words in a sentence are swapped and the goal is to detect if a model is sensitive to legal word orders.

- TreeDepth (Syntactic) tests whether a model can infer the depth of the syntactic tree of sentences.

- TopConst (Syntactic) tests whether a model can recognize the top constituents of the sentence, e.g., "*[Then] [very dark gray letters on a black screen] [appeared] [.]*" has top constituent sequence: "ADVP NP VP ". This dataset is first introduced by Shi et al. (2016).

- Tense (Semantic) asks for the tense of the main clause verb.

- SubjNum (Semantic) focuses on the number of the subject of the main clause.

- ObjNum (Semantic) tests for the number of the direct object of the main clause.

- SOMO (Semantic) checks the sensitivity of a model to random replacement of a noun or verb.

- CoordInv (Semantic) tests whether a model can recognize the order of clauses is inverted.

Figure A2 gives the results. We first observe that the combination of contextual and positional encodings can have better performances across all probing tasks (yellow lines). Secondly, compared to contextual encodings, positional encodings perform better on syntactic tasks (TreeDepth, TopConst, BShift), which require more information of word orders. On semantic tasks, contextual encodings outperform positional encodings on Tense and ObjNum while performing poorly when the semantic probing tasks require order information (CoordInv). Thirdly, a hierarchical structure exists here when we check the peak of probing tasks for

| Relation | Distance | *contextual* | *positional* | *both* |
|---|---|---|---|---|
| Root | 0.0 | 99.3 | 3.8 | 86.5 |
| auxpass | 1.1 | 44.6 | 91.1 | 92.9 |
| compound | 1.2 | 21.7 | 75.0 | 70.6 |
| aux | 1.3 | 25.2 | 77.9 | 79.1 |
| nummod | 1.3 | 26.8 | 78.9 | 75.5 |
| amod | 1.4 | 19.7 | 69.3 | 66.9 |
| det | 1.8 | 47.9 | 52.9 | 51.6 |
| advmod | 2.1 | 16.5 | 62.4 | 58.7 |
| pobj | 2.3 | 9.0 | 33.9 | 46.3 |
| nsubj | 2.3 | 13.4 | 58.2 | 52.6 |
| poss | 2.3 | 15.9 | 31.7 | 43.5 |
| dobj | 2.5 | 20.0 | 34.8 | 41.6 |
| prep | 2.6 | 23.1 | 62.8 | 63.4 |
| npadvmod | 3.1 | 14.4 | 30.0 | 43.8 |
| cc | 3.1 | 28.4 | 52.0 | 51.6 |
| mark | 4.1 | 25.1 | 21.3 | 64.4 |
| conj | 4.9 | 25.1 | 31.2 | 33.6 |
| punct | 7.3 | 25.9 | 30.3 | 32.7 |
| advcl | 7.6 | 18.4 | 9.8 | 20.1 |
| ccomp | 8.1 | 29.0 | 12.4 | 33.2 |
| short | $\leq 4$ | 28.4 | 54.3 | 61.6 |
| long | $> 4$ | 24.7 | 21.0 | 36.8 |
| Macro Avg | - | 27.5 | 46.0 | 55.4 |

Table A3: Evaluations of predictions of dependency relations on MRPC dataset. The top 20 common relations are shown. The distinction of *"short"* and *"long"* is whether the average length of the relation is greater than 4.

each model, as observed by Jawahar et al. (2019). For surface tasks, the surface knowledge is stored more in the bottom layer, syntactic knowledge is in the middle layer and semantic knowledge is in the middle and top layer. Therefore, we conclude that positional encodings play more of a role at the syntactic level tasks. On semantic tasks, especially position-independent ones, contextual encodings are more important.

### A.6.2 Dependency Analysis of Positional Encodings

The detailed scores of each relation are shown in Table A3. We find that positional attentional heads outperform contextual heads on short-distance relations, e.g., auxpass and compound. Contextual attention can capture better long-distance relations than positional attention while contextual attention itself has a certain degree of ability to detect some long-distance relations such as conj and punct. Note that there exists a head in contextual attention maps attending the token itself, therefore, the score on the Root relation is the best.

### A.7 Visualizations of Positional Weights

In Figure 1, we visualize the averaged positional weights of various pre-trained language models and identify they have similar visualized results. However, We find that the behavior of positional encodings is very diverse across attention heads. Note that there are 144 attentional heads (12 layers

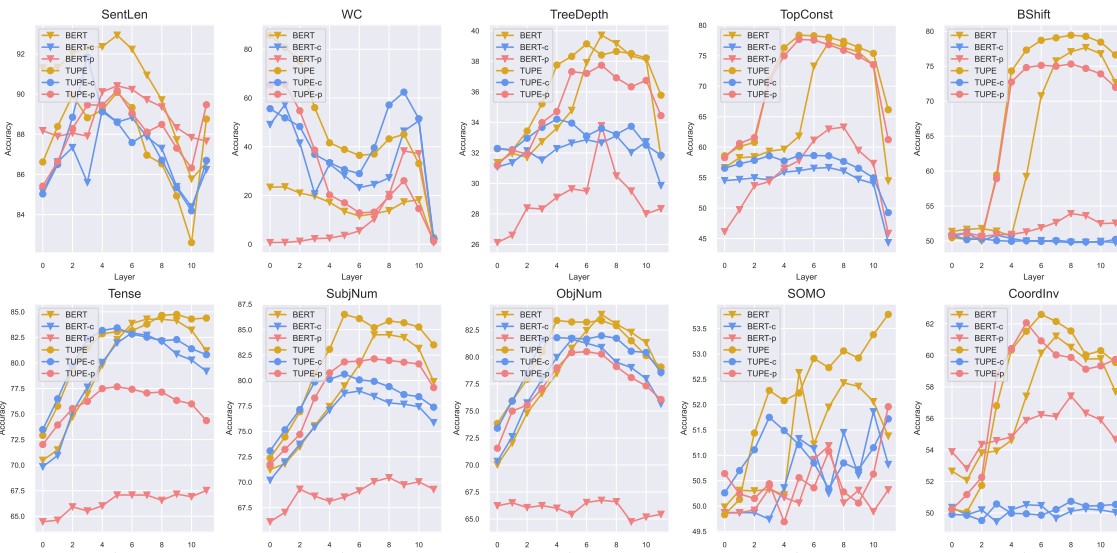

Figure A2: Results of linguistic probing tasks across different layers. BERT-based models are shown in a triangle while TUPE-based models are shown in a circle. The red, blue and yellow lines represent the use of positional weights, contextual weights and a combination of both, respectively.

$\times$ 12 heads) for the BERT$_{base}$ model. For example, the visualizations of BERT (Figure A3) validate this phenomenon. Besides, we observe that BERT exhibits a hierarchical structure: positional weights of lower layers are nearly uniform (Layer-4), middle layers attend more to local units (Layer-7) and higher layers demonstrate the asymmetric property (Layer-12). We also visualize all the positional heads in BERT-A$^*$ (Figure A4).

**Visualization Analysis of BERT-A$^*$.** BERT-A$^*$ outperforms BERT by 3.1 percentage points on average across 10 downstream tasks. The main difference between BERT-A$^*$ and BERT is the learnable handcrafted positional encodings. For visualizations, we take the positional weight $\delta_{i,j}$ in Equation 4 instead of Identical Word Probing. Figure A4 shows that most positional heads perfectly satisfy the properties of locality and symmetry, which can bring better inductive bias for sentence representations. Another observation is that the diagonal bandwidths are diverse across positional heads after learning, which means proximity units can be combined at different distances. We conclude that, compared to randomly initialized positional encodings, the encodings initialized with locality and symmetry properties lead to better sentence representation models.

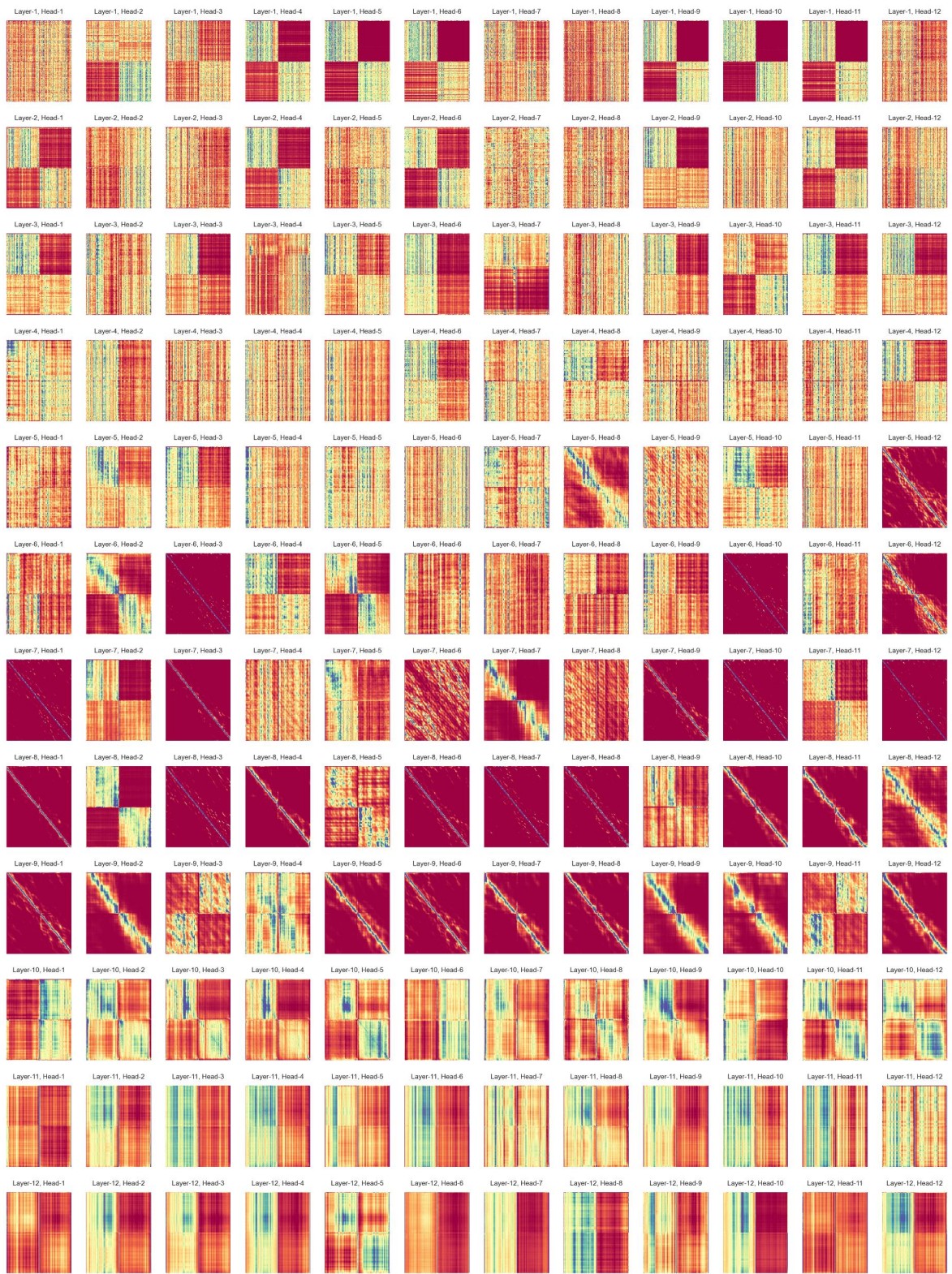

Figure A3: Visualizations of positional weights of BERT across all layers. The weights are computed by Identical Word Probing. Red color means lower values and blue color means higher values.

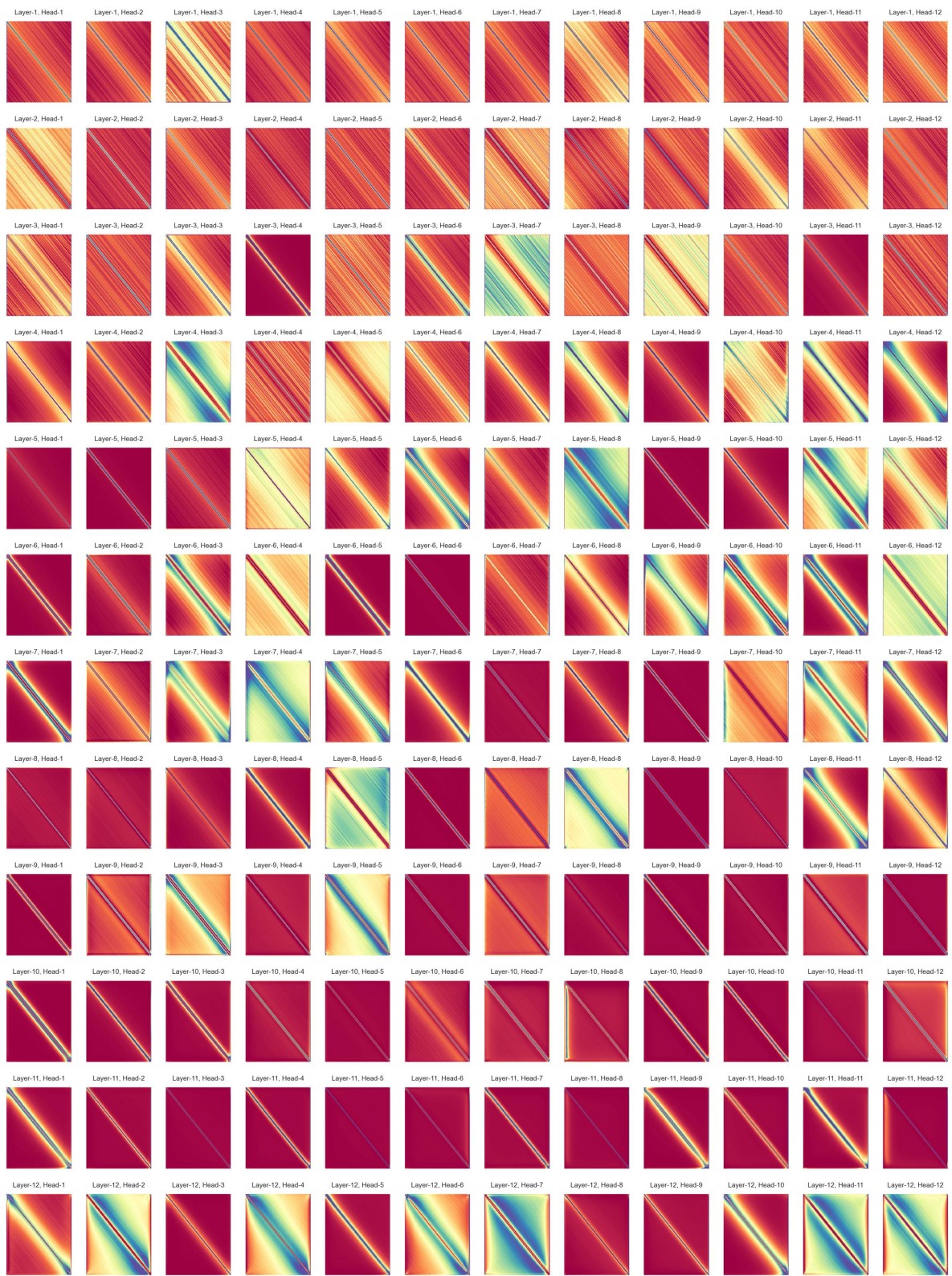

Figure A4: Visualizations of positional weights of BERT-A* across all layers. Red color means lower values and blue color means higher values.