# OpenReview forum: "The Locality and Symmetry of Positional Encodings"
_EMNLP/2023/Conference — EMNLP 2023 Findings_

### Official Review · Reviewer_R38Q · 2023-07-29

**Typos Grammar Style And Presentation Improvements:** L262
**Soundness:** 3

**Excitement:**

3: Ambivalent: It has merits (e.g., it reports state-of-the-art results, the idea is nice), but there are key weaknesses (e.g., it describes incremental work), and it can significantly benefit from another round of revision. However, I won't object to accepting it if my co-reviewers champion it.

**Paper Topic And Main Contributions:**

This work provides a detailed analysis on the positional encodings in the BERT model. Two core properties, locality and symmetry, are investigated. It is discovered that models with suitable properties can lead to better inductive bias and better performance through both non-pretraining and pretraining settings. Finally, two probing tasks are designed, showing a weakness of the encodings with the symmetry property, which may lead to insensitivity to semantic role swapping.

**Questions For The Authors:**

- A: I still do not fully understand the calculation of the Symmetry metric. For a specific entry with “j”, why is it the difference between the values with indexes of j and n-j+1? Shouldn’t “i” (the current position) also be involved here?
- B: Better designed positional encodings may also enjoy the benefits of faster convergence, and I’m wondering if such phenomenon could be observed in some of the training experiments?


**Reasons To Accept:**

- This work provides a comprehensive investigation and analysis on the positional encodings in BERT, and some of the findings seem interesting and may inspire more future analysis.
- The proposed swapping experiments seem interesting, indicating the utilization of linguistic structures in the analysis of pre-trained models.


**Reasons To Reject:**

- Only BERT-styled models are investigated, while it would be better if decoder-only and encoder-decoder models can be analyzed as well.
- The metrics are calculated using the Identical Word Probing method, which takes sequences with repeated identical words. This may be different to the real scenarios with real natural language sequences. It would be nice if more different ways for the measurement of positional encodings can be explored.


**Reproducibility:**

4: Could mostly reproduce the results, but there may be some variation because of sample variance or minor variations in their interpretation of the protocol or method.

**Reviewer Confidence:**

2: Willing to defend my evaluation, but it is fairly likely that I missed some details, didn't understand some central points, or can't be sure about the novelty of the work.

---

> ### Author Rebuttal · Authors · 2023-08-26
>
> Dear Reviewer R38Q,
>
> We appreciate the time you spent reviewing the manuscript. Also, we are encouraged that you find our analyses are comprehensive and our findings may inspire future studies.
>
> First, we would like to clarify one comment you gave in the Main Contribution part:
>
> **Q1:**
>
> >*This work provides a detailed analysis on the positional encodings in the BERT model*
>
> **A1:**
>
> In this work, we investigated various positional encodings (absolute and relative), not limited to the encodings used in BERT. For example, Figure 2 shows the visualizations of the locality and symmetry for distinct positional encodings.
>
> Next, we address your main concerns and questions.
>
> **Q2:**
>
> >  *Only BERT-styled models are investigated, while it would be better if decoder-only and encoder-decoder models can be analyzed as well*
>
> **A2:**
>
> Only Bidirectional Masked Language Models are included in this work because there is an obvious distinction between them and Decoder-only models.
>
> Decoder-only models like GPT predict the next word based on previous words in the sentence (Left-to-Right mode), and therefore implicitly introduce directional information,  which violates the symmetry property. While GPT and LLaMA have attracted a lot of attention, bidirectional language models are still widely used today, e.g., sentence embeddings [1,2], Named Entity Recognition [3, 4] and Retrieval-based Models [5,6]. Therefore, the research of their positional encodings is valuable.
>
> **Q3:**
>
> > *The metrics are calculated using the Identical Word Probing method, which takes sequences with repeated identical words. This may be different to the real scenarios with real natural language sequences. It would be nice if more different ways for the measurement of positional encodings can be explored.*
>
> **A3:**
>
> The Identical Word Probing is used only for visualizations (Figure 1 and 3), and the two metrics of our designed positional encodings can be computed directly, as described in Section 3.3. Hence, the experimental results in Section 3.3.1 and 3.3.2 are unrelated to Identical Word Probing and we reported results across multiple real natural language datasets.
>
> More importantly, using identical word probing is possible to measure the behavior of positional encodings in Transformers.
> We theoretically demonstrate that the attentional weight $\alpha_{i,j}$ can be divided into two components (in Equation 4): contextual correlations $\gamma_{i,j}$ and positional correlations $\delta_{i,j}$.  We feed many randomly selected identical words and average $\alpha$ over many words, then $\alpha$ is only related to $\delta_{i,j}$ and can reflect general behaviors of positional encodings.
>
> **Q4:**
>
> >*I still do not fully understand the calculation of the Symmetry metric. For a specific entry with “j”, why is it the difference between the values with indexes of j and n-j+1? Shouldn’t “i” (the current position) also be involved here?*
>
> **A4:**
>
> We made a mistake in Equation 8 and it does not match the code used in our experiments.
> We apologize for this.
>
> For understanding, we provided the code for calculating the symmetry of a matrix, and we average values of all vectors as the matrix-level symmetry, which reflects a general pattern.
> The current position is not involved since we aim to measure whether the positional weight vector is symmetrical to the left and right at the current position $i$.
>
> The core idea of this metric is to measure whether a positional weight vector $\epsilon_i$ is symmetrical to the left and right at the current position $i$.
> For example, we have a sentence *"cats eat mice"*  and a positional weight vector for the position of *"eat"*, $[0.25, 0.5,0.25]$, which is generated by a particular encoding. Then the vector perfectly satisfies symmetry. However, we would like to argue that the encoding is insensitive to the change of word orders like *"mice eat cats"*.
>
> ```
> def cal_matrix_symmetry(M: List[List[float]]) -> float:
>     seq_length = len(M)
>     sum_symmetry = list()
>     for index in range(seq_length):
>         left, right = M[index][0:index], M[index][index + 1:seq_length]
>         l_len, r_len = len(left), len(right)
>         min_seg_len = min(l_len, r_len)
>         if min_seg_len == 0: continue
>         left_seq, right_seq = left[-min_seg_len:], list(reversed(right[:min_seg_len]))
>         vec_sym = np.array([abs(left_seq[j] - right_seq[j]) for j in range(len(left_seq))]).reshape(-1, 1)
>         normalized_vec_sym = min_max_scaler.fit_transform(vec_sym)
>         sum_symmetry.extend([v[0] for v in normalized_vec_sym])
>     symmetry = 1 - sum(sum_symmetry) / len(sum_symmetry)
>     return symmetry
> ```
>
>
> **Q5:**
> > *Better designed positional encodings may also enjoy the benefits of faster convergence, and I’m wondering if such phenomenon could be observed in some of the training experiments?*
>
> **A5:**
>
> We have done such an experiment. Specifically, we recorded loss values of training and validation and observed that a BERT model equipped with our hand-crafted positional encodings can consistently achieve lower training and validation loss than the original BERT using the same steps, which proves that better designed PEs can have better convergence speed.
>
> The table below shows training loss values of the original BERT and BERT-A* with better initialized positional encodings.
>
> |  Model   | 5000|10000|15000|20000|25000 (steps)|
> |  ----  | ----  | ----  | ----  | ----  | ----  |
> | BERT  | 7.4 | 7.2 |7.1 |7.0 |7.0 |
> | BERT-A*  | 7.2 |6.9 |6.8 |6.7 |6.6 |
>
>
>
> [1] WhitenedCSE: Whitening-based Contrastive Learning of Sentence Embeddings. ACL 23
>
> [2] Composition-contrastive Learning for Sentence Embeddings. ACL 23
>
> [3] An Embarrassingly Easy but Strong Baseline for Nested Named Entity Recognition. ACL 23
>
> [4] Focusing, Bridging and Prompting for Few-shot Nested Named Entity Recognition. ACL Findings 23
>
> [5] RECAP: Retrieval-Enhanced Context-Aware Prefix Encoder for Personalized Dialogue Response Generation. ACL 23
>
> [6] Retrieval-Based Transformer for Table Augmentation. ACL Findings 23

---

### Official Review · Reviewer_ftCy · 2023-08-04

**Soundness:** 3

**Excitement:**

3: Ambivalent: It has merits (e.g., it reports state-of-the-art results, the idea is nice), but there are key weaknesses (e.g., it describes incremental work), and it can significantly benefit from another round of revision. However, I won't object to accepting it if my co-reviewers champion it.

**Missing References:**

-

**Paper Topic And Main Contributions:**

The paper investigates positional encodings for transformer-based LMs. It suggests two properties, symmetry and locality, as important properties for analyzing the influence of positional encodings.
1. by calculating the symmetry and locality of different LMs with learnable PEs, it is shown that not only are the symmetry and locality learned during pre-training, but that these values are also similar to those of fixed PEs.
2. A function is presented to create PEs with adjustable locality and symmetry values, which is then used to examine the effect of both properties on different NLU tasks (such as Sentiment Analysis) on different datasets. For some datasets, significant improvements are reported by tuning the PE initialization.
3. Two probing tasks of word-swaps are proposed to study the sensitivity of LMs wrt. word swaps. The first variant swaps words in a phrase of a sentence (bad syntax, semantics tends to be preserved), the second variant swaps agent and patient phrases in a sentence (semantics tends to be changed). By shuffling and evaluating on SNLI, it is shown that LMs can handle the first variant but perform poorly on the second.

**Questions For The Authors:**

A. I’m missing a more detailed investigation if really symmetry is the reason that the models perform poorly on the SR probing task. Can't there be other reasons for the observed effect? Maybe I'm misunderstanding, but in Section 3.4 you even claim that PEs matter more at the syntactic level.

B. I also had a look at the code: In classify.py, the if statements for fp / fn / tn (starting in line 364) for one metric (MCC on COLA) appear to be incorrect. This is a major concern if this code was used to compute the values presented in the paper. Can you clarify?


**Reasons To Accept:**

- Research of PEs is an interesting topic and relevant to EMNLP.

- The paper includes an extensive analysis of positional encodings on different NLU tasks. It also shows that initializing BERT with symmetric / localized PEs can significantly improve results on some NLU tasks.

- The proposed probing tasks seem useful to further investigate the effect of word swaps.


**Reasons To Reject:**

Eq. 7 and 8 define symmetry, which is one of the key concepts of the paper. However, the equations do not appear to match the original symmetry as proposed by Wang et al.. I'm not convinced that these equations measure proper symmetry of the attention matrix (as only values of a single attention weight vector are used).

For the results reported in the non-pre-training setting (Section 3.5.1.), improvements in accuracy are rather small (< 2 pp).



**Reproducibility:**

3: Could reproduce the results with some difficulty. The settings of parameters are underspecified or subjectively determined; the training/evaluation data are not widely available.

**Reviewer Confidence:**

3: Pretty sure, but there's a chance I missed something. Although I have a good feel for this area in general, I did not carefully check the paper's details, e.g., the math, experimental design, or novelty.

**Typos Grammar Style And Presentation Improvements:**

- Only two variants are compared with the original BERT (line 352)

- Theta is not part of Eq. 4 (or any other equation in the paper) (line 290, 292)

- “electicity guitar” → “electric guitar” (line 518)

- Some sentences are incomplete (e.g. line 262, 582)

---

> ### Author Rebuttal · Authors · 2023-08-27
>
> Dear Reviewer ftCy,
>
> We would like to thank you for your time and detailed feedback. We are happy that you found our research topic interesting and our proposed probing tasks useful for future studies.
>
> **Q1:**
>
> >*Eq. 7 and 8 define symmetry, which is one of the key concepts of the paper. However, the equations do not appear to match the original symmetry as proposed by Wang et al.. I'm not convinced that these equations measure proper symmetry of the attention matrix (as only values of a single attention weight vector are used).*
>
> **A1:**
>
> Thank you for pointing us to this issue. Equation 7 and 8 is indeed stated incorrectly in the paper. We apologize for this. The code, though, is correct (attached below for your reference).
> Our symmetric metric is different from the one proposed by Wang et al.
> because we aim to measure whether a positional weight vector $\epsilon_i$ is symmetrical to the left and right at the current position $i$, and the original metric is used to check the symmetry of a matrix.
> For example, we have a sentence *"cats eat mice"*  and a positional weight vector for the position of *"eat"*, $[0.25, 0.5,0.25]$, which is generated by a particular encoding. Then the vector perfectly satisfies symmetry. However, we would like to argue that the encoding is insensitive to the change of word orders like *"mice eat cats"*.
>
> Below is the code for calculating the symmetry of a matrix, and we average values of all vectors as the matrix-level symmetry, which reflects a general pattern of PEs.
> ```
> def cal_matrix_symmetry(M: List[List[float]]) -> float:
>     seq_length = len(M)
>     sum_symmetry = list()
>     for index in range(seq_length):
>         left, right = M[index][0:index], M[index][index + 1:seq_length]
>         l_len, r_len = len(left), len(right)
>         min_seg_len = min(l_len, r_len)
>         if min_seg_len == 0: continue
>         left_seq, right_seq = left[-min_seg_len:], list(reversed(right[:min_seg_len]))
>         vec_sym = np.array([abs(left_seq[j] - right_seq[j]) for j in range(len(left_seq))]).reshape(-1, 1)
>         normalized_vec_sym = min_max_scaler.fit_transform(vec_sym)
>         sum_symmetry.extend([v[0] for v in normalized_vec_sym])
>     symmetry = 1 - sum(sum_symmetry) / len(sum_symmetry)
>     return symmetry
> ```
>
> **Q2:**
>
> >*For the results reported in the non-pre-training setting (Section 3.3.1.), improvements in accuracy are rather small (< 2 pp).*
>
> **A2:**
>
> In this experiment, our goal is to validate there is a correlation between our proposed two metrics and performances of downstream tasks, and we do not claim that our positional encodings can bring any improvements.
>
> Moreover, we accounted for variance in this experiment and the results are statistically significant. As described in line 310, each model is trained for 5 epochs and we select the best model on validation sets to evaluate on the test set. We repeat this procedure 5 times and use the average score to report.
> To draw Figure 3, we repeat this procedure ~100 times to obtain the accuracy of models with different locality and symmetry values.
>
>
> **Q3:**
>
> >*I’m missing a more detailed investigation if really symmetry is the reason that the models perform poorly on the SR probing task. Can't there be other reasons for the observed effect? Maybe I'm misunderstanding, but in Section 3.4 you even claim that PEs matter more at the syntactic level.*
>
> **A3:**
>
> The symmetry property is one of the main causes of the effect.
> As the example in A1, we have a sentence *"cats eat mice"*  and a positional weight vector for the position of *"eat"*, $[0.25, 0.5,0.25]$, which is generated by a particular encoding. Then the vector perfectly satisfies symmetry. However, the encoding is insensitive to the change of word orders like *"mice eat cats"*.
> Another reason is that most existing pre-training tasks like masked token prediction do not explicitly require positional information. In Table 3, the better performance of StructBERT on the Shuffle-SR suggests that introducing additional order-sensitive training tasks may improve this behavior since it explicitly models language structures by forcing it to reconstruct the right order of words during pre-training.
>
> Our empirical results show that there is a clear distinct role between the positional
> and contextual encodings in sentence comprehension: positional encodings play more of a role at the
> syntactic level while contextual encodings serve more at the semantic level (if the semantic task does
> not require word order information), as shown in Figure A2 in the appendix.
>
> **Q4:**
>
> > *I also had a look at the code: In classify.py, the if statements for fp / fn / tn (starting in line 364) for one metric (MCC on COLA) appear to be incorrect. This is a major concern if this code was used to compute the values presented in the paper. Can you clarify?*
>
> **A4:**
>
> We agree with your statement, but the CoLA task is not included in our experiment.
> We appreciate the time you spent checking our code carefully, and we will use it to correct this piece of code even if it is not currently used.

---

### Official Review · Reviewer_5QeA · 2023-08-11

**Soundness:** 4

**Excitement:**

3: Ambivalent: It has merits (e.g., it reports state-of-the-art results, the idea is nice), but there are key weaknesses (e.g., it describes incremental work), and it can significantly benefit from another round of revision. However, I won't object to accepting it if my co-reviewers champion it.

**Paper Topic And Main Contributions:**

The authors argue that position encodings have two main properties, Locality and Symmetry. They also propose novel probing tasks to quantitatively study these two properties of fixed or pre-trained position encodings. They further reveal existing pre-trained language models are robust to process constituency-shuffling texts but fail to represent semantic-role-shuffling ones.

**Reasons To Accept:**

1. It is interesting to study the Symmetry property of position encodings and to discuss its drawback.
2. The paper contributes new probing tasks of shuffle words, and provides experimental analysis to prove models can be robust against local swaps but sensitive to global swaps.
3. The authors examine several encoding methods and many pre-trained models, which provides convincing conclusions.

**Reasons To Reject:**

1. The experiments are merely implemented with BERT-style encoder-only models, whose scales are no larger than BERT-large. The contribution of this work is limited, since many nowadays popular models are large decoder ones like GPT-3.5 and LLaMA.
2. The contributions of *pre-training* and *design of position encodings* are not clear. As mentioned in L379-383, learned encodings may not have the best locality and symmetry, but they can still represent contexts. There might be non-positional information but implicitly indicates the positions of tokens, for example, commonsense. We probably say a dog is biting a person, but rarely say a person biting a dog. It might affect the model performance of downstream tasks like NLI, since models can inference merely by part of keywords.

**Reproducibility:**

4: Could mostly reproduce the results, but there may be some variation because of sample variance or minor variations in their interpretation of the protocol or method.

**Reviewer Confidence:**

3: Pretty sure, but there's a chance I missed something. Although I have a good feel for this area in general, I did not carefully check the paper's details, e.g., the math, experimental design, or novelty.

---

> ### Author Rebuttal · Authors · 2023-08-27
>
> Dear Reviewer 5QeA,
>
> We appreciate that you found our work interesting and our experiments provided convincing conclusions. We also thank you for your critical comments.
>
> **Q1:**
>
> > *The experiments are merely implemented with BERT-style encoder-only models, whose scales are no larger than BERT-large. The contribution of this work is limited, since many nowadays popular models are large decoder ones like GPT-3.5 and LLaMA.*
>
> **A1:**
>
> Only Bidirectional Masked Language Models are included in this work because there is an obvious distinction between them and Decoder-only models.
>
> Decoder-only models like GPT predict the next word based on previous words in the sentence (Left-to-Right mode), and therefore implicitly introduce directional information,  which violates the symmetry property.
> Furthermore, recent studies have shown that decoder-only transformers without positional encodings are able to achieve competitive or even better performance than other explicit positional encoding methods [1, 2], which shows the distinction of decoder-only models.
>
> While GPT and LLaMA have attracted a lot of attention, bidirectional language models are still widely used today,  e.g., for sentence embeddings [4,5], Named Entity Recognition [6, 7] and Retrieval-based Models [8,9]. Therefore, the research of their positional encodings is valuable.
>
>
> **Q2:**
>
> > *The contributions of pre-training and design of position encodings are not clear. As mentioned in L379-383, learned encodings may not have the best locality and symmetry, but they can still represent contexts. There might be non-positional information but implicitly indicates the positions of tokens, for example, commonsense. We probably say a dog is biting a person, but rarely say a person biting a dog. It might affect the model performance of downstream tasks like NLI, since models can inference merely by part of keywords.*
>
> **A2:**
>
> Thank you for raising this interesting question.
> To study the effect of varying locality and symmetry, we designed the attenuated encoding that can be parameterized along the two metrics. We are able to observe the performance of a model at a particular locality and symmetry value. In the pre-training setting experiment, a model is initialized with locality and symmetry that have the same values as the learned positional encodings in BERT and the results in Table 1 validate that better locality and symmetry can yield better inductive bias.
>
> As the example you gave, we agree that there exists some implicit position information, e.g., the special token [CLS], which always appears in the first position and can be regarded as a reference point to estimate positions. However, the usage of positional encodings is essential for transformer models, as shown by prior studies [2,3]
>
> [1] The impact of positional encoding on length generalization in transformers. arXiv 2023
>
> [2] What Language Model to Train if You Have One Million GPU Hours?. EMNLP Findings 2022
>
> [3] Constituency Parsing with a Self-Attentive Encoder. ACL 2018
>
> [4] WhitenedCSE: Whitening-based Contrastive Learning of Sentence Embeddings. ACL 23
>
> [5] Composition-contrastive Learning for Sentence Embeddings. ACL 23
>
> [6] An Embarrassingly Easy but Strong Baseline for Nested Named Entity Recognition. ACL 23
>
> [7] Focusing, Bridging and Prompting for Few-shot Nested Named Entity Recognition. ACL Findings 23
>
> [8] RECAP: Retrieval-Enhanced Context-Aware Prefix Encoder for Personalized Dialogue Response Generation. ACL 23
>
> [9] Retrieval-Based Transformer for Table Augmentation. ACL Findings 23

---

### Meta-Review · Area_Chair_GkzS · 2023-09-29

**Recommendation:** 3

**Metareview:**

Paper studies positional encodings of encoder only models for NLP (BERT) and proposes probing tasks and formulation to investigate symetry and locality of the positional encodings. Reviewers raised some concerns about the experimental design and formulation.

---

### Decision · Program_Chairs · 2023-10-07

**Decision:**

Accept-Findings

**Comment:**

Paper studies positional encodings of encoder only models for NLP (BERT) and proposes probing tasks and formulation to investigate symetry and locality of the positional encodings. Reviewers raised some concerns about the experimental design and formulation.